# Involvement of the H3.3 Histone Variant in the Epigenetic Regulation of Gene Expression in the Nervous System, in Both Physiological and Pathological Conditions

**DOI:** 10.3390/ijms241311028

**Published:** 2023-07-03

**Authors:** Carlo Maria Di Liegro, Gabriella Schiera, Giuseppe Schirò, Italia Di Liegro

**Affiliations:** 1Department of Biological, Chemical and Pharmaceutical Sciences and Technologies (STEBICEF), University of Palermo, 90128 Palermo, Italy; carlomaria.diliegro@unipa.it (C.M.D.L.); gabriella.schiera@unipa.it (G.S.); 2Department of Biomedicine, Neurosciences and Advanced Diagnostics (Bi.N.D.), University of Palermo, 90127 Palermo, Italy; giuseppe.schiro@you.unipa.it

**Keywords:** histone variants, H3.3 histone, histone post-translational modifications (PTMs), epigenetic modification of gene expression in brain development and function

## Abstract

All the cells of an organism contain the same genome. However, each cell expresses only a minor fraction of its potential and, in particular, the genes encoding the proteins necessary for basal metabolism and the proteins responsible for its specific phenotype. The ability to use only the right and necessary genes involved in specific functions depends on the structural organization of the nuclear chromatin, which in turn depends on the epigenetic history of each cell, which is stored in the form of a collection of DNA and protein modifications. Among these modifications, DNA methylation and many kinds of post-translational modifications of histones play a key role in organizing the complex indexing of usable genes. In addition, non-canonical histone proteins (also known as histone variants), the synthesis of which is not directly linked with DNA replication, are used to mark specific regions of the genome. Here, we will discuss the role of the H3.3 histone variant, with particular attention to its loading into chromatin in the mammalian nervous system, both in physiological and pathological conditions. Indeed, chromatin modifications that mark cell memory seem to be of special importance for the cells involved in the complex processes of learning and memory.

## 1. Introduction

Since Gurdon’s experiments [1], based on the transplantation into enucleated oocytes of nuclei purified from somatic cells, it became clear that during differentiation, cells do not lose DNA and thus maintain a nucleus with intact potential to generate an entire organism. Despite the presence of the entire genome, however, each specialized cell expresses only a very small percentage of its genome. The reason for specific gene selection lies in the tridimensional organization of chromatin, which is a complex of DNA and proteins. Among these latter molecules, the most represented are histones, i.e., basic proteins that are highly conserved in evolution, which interact with DNA, allowing condensation of the nuclear genome in the very small volume of eukaryotic nuclei. The first level of chromatin organization is the nucleosome, in which about 147 base pairs (bp) of DNA are wrapped around a protein octamer formed by two molecules of each of the histones H2A, H2B, H3, and H4 (called “core” histones”). A fifth histone, the “linker” H1 histone then seals together the points at which DNA enters and exits the nucleosome. Interaction among H1 molecules allows the formation of more condensed DNA fibers [2,3,4,5,6,7,8].

Most importantly, the chromatin structure differs at the level of different genes that, depending on their structural organization, can be transcriptionally repressed or active, and this is why different cells are able to express different and specific genes, as well as a family of common genes involved in basic metabolism. Thus, specific properties and the behavior of somatic cells do not depend on changes in the genotype but on the specific arrangement of chromatin, according to the concept of “epigenetics” proposed by Waddington many decades ago [9], which is now widely accepted [4,10].

The chromatin structure is highly dynamic and changes in different cells during development and differentiation; it also changes in terminally differentiated cells in response to specific inducing factors, such as, for example, thyroid or steroid hormones [11]. At least three connected mechanisms are known to induce structural rearrangements of chromatin (Figure 1): (i) post-translational modification (PTM) of histone proteins [10,12] together with DNA methylation [10,13,14]; (ii) the activity of ATP-dependent complexes that are able to induce modifications in the structure/position of nucleosomes [15], and, finally, (iii) the synthesis and incorporation of histone variants into chromatin [7,16,17,18].

Herein, we will mainly focus on the latter mechanism (Figure 1, pathway c) and, in particular, on the specific involvement of the H3 histone variant, known as H3.3 core histone, in the epigenetic regulation of gene expression in the nervous system during maturation, as well as in the acquisition of complex functions such as learning and memory. We also discuss the involvement of H3.3 in brain cancer.

## 2. General Properties of Genes Encoding Histone Variants

Although highly conserved in evolution, and thus among different species, histone proteins exist under different isoforms in the same species. These isoforms seem to have different effects on chromatin structure and then on gene expression.

Interestingly, genes encoding the main histone species differ both in structure and expression from those encoding variants synthesized only in specific moments of cell differentiation. In particular, the main histone species (also known as replication-dependent or canonical histones) are synthesized exclusively during the S phase of the cell cycle when DNA is replicated. The corresponding genes are highly repeated and very often clustered; moreover, they do not contain introns, and the corresponding mRNAs are not poly-adenylated [19,20,21]. All these features are related to the necessity of having mRNAs that are immediately available for translation and then for degradation. On the other hand, genes encoding constitutive/variant histones (also known as replication-independent or non-canonical histones) are similar to all the other genes in that they are mostly unique genes that are transcribed, independently of DNA replication, into mRNAs that can contain introns and are polyadenylated [22]. These genes should be regulated during differentiation in order to produce proteins that are able to bind specific regions of chromatin, thus allowing activation/repression of specific genes.

Many years ago, our group showed that two histone variants, i.e., the linker histone H1.0 and the core histone H3.3, are specifically expressed in the rat brain during brain maturation [23]. Interestingly, a combination of run-on experiments on isolated nuclei and transcription inhibition using actinomycin D demonstrated that the two genes have an “open” structure and that H1.0 and H3.3 histone synthesis in the central nervous system (CNS) is largely regulated at the post-transcriptional level [24]. Indeed, we identified a group of proteins that are able to bind their mRNAs and cloned a couple of them (CSD-C2/PIPPin and LPI/PEP-19) [25,26,27,28,29]. Notably, in a very recent and interesting paper focused on the effects of hunger on neuronal histone modifications and the life span of the *Drosophila* fruit fly, the effects of a diet containing low amounts of branched-chain amino acids (BCAAs) were analyzed. In particular, the authors found that total histone H3 abundance decreased in flies fed a low BCAA diet, while H3 mRNA increased [30]. This observation suggested that post-transcriptional events might be of importance for general H3 metabolism. Moreover, the paper also reported that canonical H3 is evicted from chromatin and replaced with H3.3 [30].

It is also worth noting that two genes encoding the H3.3 histone are present in mammals: *H3.3A* and *H3.3B* (also called *H3F3A* and *H3F3B*), which are located on different chromosomes. While the distribution of exons and introns, as well as promoters and other regulatory regions, are different in the two genes, the corresponding proteins are identical [31] and also highly conserved in evolution (see Figure 2). Thus, it is highly probable that the existence of the two genes is not important for having two proteins with different activities but, instead, because it offers the possibility to regulate the genes (and the related mRNAs) independently and/or with different mechanisms [32]. In particular, it was suggested that the two genes may have cell type-specific expression [33], although their overall activity in different tissues was reported to be quite similar [34].

Actually, the H3.3 protein is not so different with respect to the canonical H3.1 and H3.2 isoforms: indeed, it differs by only five and four amino acids, respectively, from them [35,36,37]. However, these amino acids are, for example, sufficient to allow H3.3 to interact with specific histone chaperones, such as the Death domain-associated protein (DAXX), the alpha-thalassemia/mental retardation X-linked protein (ATRX) complex, and the histone regulator A (Hira)/calcineurin-binding protein 1 (Cabin 1)/ubinuclein1 (Ubn1) complex, involved in its loading on chromatin [7,38,39,40,41]. The H3.3 interaction with these chaperones is determinant for its deposition on specific regions of the genome. It was shown, for example, that a mutation of the ATRX complex leads, as a consequence, to variation in the deposition of H3.3 and chromatin accessibility in association with an alteration in gene expression [42].

The H3.3 variant is indeed very often bound to active chromatin and regulates transcription. In 2002, Ahmad and Henikoff reported that in *Drosophila* cells, H3.3 is the only H3 species deposited in chromatin in a replication-independent way, and they suggested that this event might be responsible for the activation of genes previously silenced because of histone PTMs [43]. As a confirmation of its gene-activating function, the H3.3 histone was also found at the level of active enhancers [44,45,46].

It was also reported that some chromatin remodeling complexes, such as the SWItch/Sucrose Non-Fermentable (SWI/SNF) complex and, in particular, its subunit “T-rich interactive domain-containing protein 1A” (ARAD1A) are required for maintaining the H3.3 histone at the level of regulatory sequences, among which are the so-called super-enhancers [47]. On the other hand, by interacting with the ATRX/DAXX chaperones, H3.3 might also be loaded on pericentric heterochromatin and telomeres [38,48,49]. This event, together with lysine 56 (H3.3K56) acetylation, seems necessary for chromosome segregation in mammals. Indeed, it was shown that cell lines carrying the mutation K56R increase cell death and modify cell morphology [50].

Recently, a brain-specific function of the chromodomain-helicase-DNA binding protein 1 (CHD1) which is a member of the SWI/SNF family of chromatin remodeling complexes was reported in *Drosophila*. CHD1 is indeed involved in the loading of H3.3 in the fly brain, where it seems to contribute to the regulation of genes that control the homeostasis of hunger and satiety signals [51]. On the other hand, as a demonstration of the wide range of tissues and functions in which H3.3 histone is probably involved, it was also found to be essential for the chromatin transitions that accompany *Drosophila* male germline maturation [52].

Further work is required to understand how H3.3, which, as mentioned above, is not so different from the canonical H3 species, can stimulate transcription. Evidently, its sequence should contain features that are able to attract, directly or indirectly, the transcriptional apparatus to the genes to which it is bound. Interestingly, for example, the H3.3 amino-terminal tail contains a serine residue (S31) that is not present in the other H3 species (which contain, instead, an alanine at that position). Moreover, this serine can be phosphorylated, and it has been suggested that this might represent a feature determinant for preferential transcription [53]. It should be noted that S31 is actually S32 in the original amino acid sequence of the H3.3 protein (Figure 2); however, in the mature protein, it becomes S31 because the initiator methionine is immediately cleaved during translation ([54] and references therein). Notably, it was recently demonstrated that S31 phosphorylation can also modify the accessibility of regulatory factors at telomeres during replication, thus stabilizing heterochromatin probably by influencing the activity of histone lysine demethylase 4B (KDM4B) [55].

One possible link between H3.3 modification and gene expression was suggested by Martire and co-workers [56]. Using mouse embryonal stem cells (mESCs), these authors showed that the cells missing histone H3.3 cannot normally acetylate the enhancers that are activated during differentiation and, more specifically, show a reduction in the acetylation of the H3 histone at lysine 27. The normal ability to regulate acetylation would depend on the stimulation of p300 acetyltransferase by phosphorylation at specific sites of the H3.3 histone variant [56,57].

**Figure 2 ijms-24-11028-f002:**
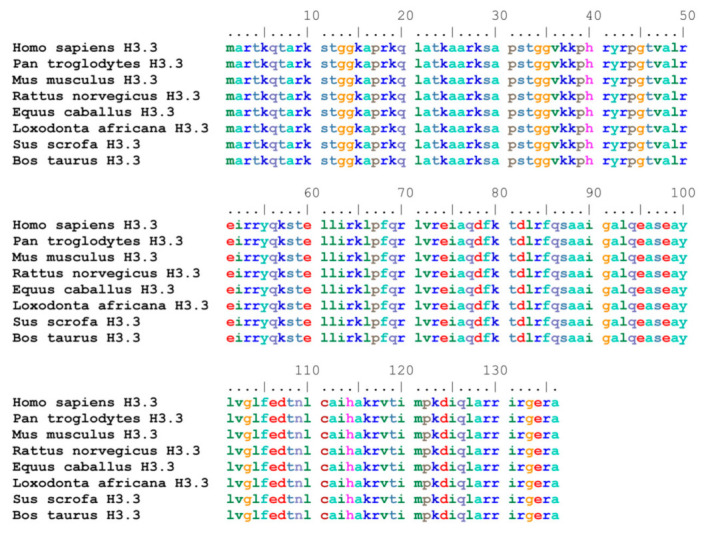
Alignment of H3.3 histones in different mammalian species. NCBI reference sequences reported: *Homo sapiens* (NP_005315.1); *Pan troglodytes* (NP_001267377.1); *Mus musculus* (NP_032237.1); *Rattus norvegicus* (NP_446437.1); *Equus caballus* (XP_023499762.1); *Loxodonta africana* (XP_010595256.1); *Sus scrofa* (NP_999095.1); *Bos taurus* (NP_001014411.1). Alignment of the shown sequences was completed using the Bioedit sequence alignment editor [58].

## 3. The H3.3 Variant in Brain Development and Maturation as Well as in the Processes of Learning and Memory

As mentioned above, all the cells of the same organism have the same genome. However, thanks to rearrangements in the structural organization of chromatin, each cell expresses only the genes necessary for basal metabolism, as well as the genes involved in its tissue-specific functions. The cell-specific overall organization of chromatin, in turn, derives from the specific history of that cell. In other words, we can state that the nuclear organization of the genome is a form of cell memory, since, at any moment, it reflects the experiences that a cell has had in its life. This epigenetic memory is built up with the specific combination and location of the different histone variants and with their specific post-translational modifications (PTMs) [59,60], thus creating a sort of “histone code” [12,61] that is superimposed to the genetic code. In the case of the H3 histone isoforms, Hake and Allis proposed the “H3 barcode hypothesis”, which suggests the existence of chromatin indexing based on the specific distribution of canonical and non-canonical H3 species, as well as on their PTMs [62]. Notably, it was also reported that H3.3 accumulates with age in most somatic tissues of mice and that this accumulation correlates with a general age-dependent change in H3 methylation [63]. This observation might be the most important since, in spite of the large body of evidence showing the involvement of H3.3 in gene transcription, it still remains not completely clear how the presence of this histone variant in chromatin can drive the activation of specific genes. Thus, it is highly probable that the H3.3 effect is also due to its specific interaction with PTM writers and/or PTM readers.

Interestingly, many chromatin-specific marks should be transmitted to the daughter cells during each cycle of DNA replication and cell division [64]; if the marks are lost, specific gene expression and, hence, central aspects of cell phenotype are equally lost. It was shown, for example, that the canonical isoforms of the H3 histone, as well as their epigenetic marks, are transferred from the mother to the daughter cells [64,65,66,67]. Similarly, H3.3-containing nucleosomes can be replicated as such during cell division, thus contributing to the epigenetic memory of the cells [68,69,70]. The importance of the events that allow transgenerational transmission of these marks is also underlined by the fact that in vertebrates, loss of chromatin assembly factor 1 (CAF1), the main chaperone involved in the DNA replication-dependent deposition of canonical H3 histones, blocks embryo development [71,72]. On the other hand, the non-canonical H3.3 histone seems to be specifically loaded at chromatin sites where nucleosomes should face rapid turnover, thus allowing active transcription [37,73,74]. Indeed, the position of H3.3 in chromatin often coincides with that of RNA polymerase II (RNAPII) [58].

As for other cells, epigenetic marks, such as histone acetylation and DNA methylation, accumulate in neuronal chromatin during cellular responses to received internal and/or environmental stimuli [75,76,77,78,79]. As a whole, chromatin DNA and protein modifications are fundamental in the control of neural progenitor (NPC) development and differentiation; for example, trimethylation of lysine 27 in the H3 histone (H3K27me3, a repressive mark) should be removed by the Jumonji C domain-containing 3 (Jmjd3) demethylase in order to allow embryonic stem cells (ESCs) to give rise to NPCs [80,81,82]. Similar events are also linked to neuronal migration [82,83]. Interestingly, in the epigenetic control of gene expression during brain maturation, microRNAs seem to also be involved, thanks to cross-talk with other epigenetic regulators. In particular, the identification of astrocyte-enriched microRNAs has been reported, and their possible role in NPC differentiation into glial cells has been suggested [84].

It is also worth noting that, recently, a new kind of histone PTM was described: modification by monoamine neurotransmitters, such as dopamine and serotonin [85,86,87,88]. Some of these modifications seem to have H3 histones as specific targets [87,88]. For example, it was reported that serotonylation involves glutamine 5 of H3 histones and that the presence of this modified H3Q5 can, in turn, stabilize H3K4 methylation [89]. While further studies are necessary to fully understand the mechanisms, functions, and regulation of histone monoamination, it is already clear that this kind of PTM can have a powerful effect on the nervous system, where neurotransmitters, such as dopamine, noradrenaline, or serotonin, are highly represented. Notably, it is known that, in addition to the monoamines stored in synaptic vesicles and ready for neurotransmission, a significant number of them are not in vesicles but are present in the neuronal soma and even in the cell nucleus. Moreover, these molecules were involved in development well before the appearance of nerve cells [88,90,91]. Notably, it was also reported that histone H3 dopaminylation can be involved in the response to abuse drugs, such as cocaine [92,93].

In addition to PTMs, loading of the H3.3 histone variant onto chromatin seems to be required for neuronal differentiation and, in particular, for activation of mature neuron-specific genes [94,95]. For example, during mouse lens fiber cell differentiation, high concentrations of H3.3 were observed at the level of transcribed regions in chromatin, like those including genes encoding crystallins, gap junction components, and intermediate filaments [96]. Actually, H3.3, together with an H2A variant known as H2A.Z, contributes to opening chromatin structure to transcription factors, by increasing accessibility of the upstream regulatory sequences present in the genes to which they are bound [57,97,98]. However, the precise mechanisms underlying how H3.3 causes chromatin opening are not completely clear. For example, using a genomic analysis on mouse embryonic stem cells in which H3.3 knockout had been induced, it was found that there is a general deregulation of promoter activity at the level of expressed genes associated with a reduction in transcription factor binding and a consequent reduction in RNA polymerase II at the transcription start sites [57].

Now, if we accept the principle that nuclear chromatin organization confers to each cell a specific phenotype because it represents the epigenetic memory of the history of that cell [99], then we have to admit that it will be even more so, and lifelong, for cells that are responsible for the macroscopic ability to learn and remember, namely neurons and probably also some glial cells such as astrocytes [100]. Indeed, epigenetic changes in chromatin, such as post-translational modifications of histones and, in particular, acetylation, are involved in the phenomenon of long-term potentiation (LTP), which is in the stabilization of memory [82,94,101,102,103]. For example, it was found that H3 and H4 histone acetylation at the level of the promoter for the gene encoding the brain-derived neurotrophic factor (BDNF) is fundamental for increasing BDNF expression in the rat hippocampus and for allowing LTP [82].

In addition to a variety of histone PTMs, the H3.3 histone is also enriched on active chromatin related to memory storing [104], and it seems to be fundamental for maintaining neuronal activity in the mouse hippocampus, as an H3.3 decrease can alter long-term memory [94]. Again, the H3.3 histone shares its ability to act as a regulator of hippocampal memory with another histone variant known as H2A1, the localization of which was found to dynamically change during the learning process [105].

As mentioned above, the H3.3 histone also has a role to play at certain heterochromatic regions, such as telomeres and pericentric repeats, where it is trimethylated at the level of its lysine 9 (H3.3K9me3), a modification bound to its ability to stabilize heterochromatin [49]. Recently, it was reported that phosphorylation at the level of its serine 31, one of the residues for which it differs from H3.1 and H3.2, stabilizes heterochromatin by inhibiting lysine-specific demethylase 4B (KDM4B), which could otherwise demethylate H3K9/K36, thus damaging the heterochromatic structure [55].

The importance of the H3.3 histone in brain development was confirmed with the discovery and analysis of different neurodevelopmental disorders. For example, mutations in the H3.3A/H3.3B genes have been indicated as new candidates for microcephaly, and intellectual disability, as well as in a novel neurodevelopmental disorder now known as Bryant-Li-Bhoj syndrome [32,106,107]. Indeed, experiments in which H3.3 expression was silenced with RNA interference demonstrated profound alterations in both the structural and functional organization of the brain, with clear effects on the mechanisms underlying memory processes [94,95]. Moreover, stage-dependent deletion of the two H3.3-encoding genes demonstrated that synthesis of H3.3 is required in the first postnatal days of rat life, thus indicating the importance of de novo accumulation of this variant at the beginning of brain development [108]. In this phase, it was clearly shown that genes involved in the proliferative stage of brain development are silenced, while a collection of genes are activated in order to drive neuronal differentiation, with even different genes involved in the specification of different layers of the developing cortex [108,109,110,111].

## 4. Histone Post-Translational Modifications and the H3.3 Variant in Neurodegenerative Diseases

Some studies have also shown that PTMs of histones can contribute to the pathogenesis of neurodegenerative diseases, such as Parkinson’s disease (PD) and Huntington’s disease (HD). In an experimental model of PD, exposure of dopaminergic cells (N27 cells) from the rat midbrain to pesticides, i.e., neurotoxic substances, induced hyperacetylation of the core histones H3/H4 and the subsequent death of dopaminergic neurons. The inhibition of hyperacetylation was neuroprotective, preventing N27 cell apoptosis [112]. Notably, it was suggested that histone hyperacetylation in PD may be a consequence of mitochondrial damage, a dysfunction-enhancing progression in PD. Indeed, in both ex vivo and in vivo neurodegenerative models of PD, induced perturbation of the mitochondria, reduced ATP production, and altered intracellular activity of proteasomes induced the hyper-activation of histone acetyltransferase (HAT) and inhibition of histone deacetylase (HDAC). This resulted in H3.3 lysine 27 hyperacetylation (H3.3K27). This epigenetic alteration directly induced the death of dopaminergic neurons by affecting gene transcription. Also, in post-mortem brains from patients with PD, histone extracts from substantia nigra lysates showed a significant nuclear accumulation of H3K27 acetylation in comparison to age-matched controls [113]. Reduced trimethylation of histone H3 lysine 4 (H3K4) on promoters of down-regulated genes was also evidenced in mice and human models of Huntington’s disease (HD). Moreover, loss of the enzyme known as little imaginal discs (Lid), a trimethyl-H3K4 demethylating enzyme, reduces neurodegeneration in *Drosophila* models of HD [114]. Alteration of H3K4 methylation was also demonstrated by Dong and collaborators using human post-mortem tissues [115]. However, modulation of histone methylation and demethylation may have opposite effects on neurodegeneration in HD. Indeed, reduced methylation of H3K27 for mutations of the polycomb repressive complex 2 (PRC2), which is responsible for the methylation of H3K27, promoted neurodegenerative processes, while inhibition of the activity of the H3K27 demethylase Utx (ubiquitously transcribed tetratricopeptide repeat, X chromosome) reduced pathological modifications in HD [116].

Interestingly, it was also suggested that mutant huntingtin, which accumulates in HD, can interact with HAT enzymes, inducing a decrease in their activity [60,117]. This observation is of special importance when we consider that histones should be acetylated in order to be degraded [118], and that epigenetic modifications of cell activity also include histone substitution at specific loci in chromatin. On the other hand, it is also important to consider that histone acetylation requires, as a substrate, Acetyl-CoA produced in the mitochondria, and this clearly means that, in general terms, metabolic dysfunctions can also hamper histone PTMs [119].

## 5. The H3.3 Variant in Cancer

Notably, alterations of histone sequences and PTMs play a role in cancer [21,120]. As discussed above, chromatin modification and accessibility largely depend on enzymatic systems that introduce PTMs, on histone variant loading into chromatin, and on factors that are able to read such modifications, thus activating transcription. Obviously, given the H3.3 transcription activating function, it does not come as a surprise that it is found mutated in many tumors [121,122,123]. In some cases, it also promotes metastasis, favoring all the mechanisms that underly the basis of this phenomenon. For example, the genes for epithelial–mesenchymal transition (EMT), like SRY-box transcription factor 9 (Sox9) and Snail, are stimulated in certain cancers [123], also thanks to the stabilization of the Hira complex [124].

Interestingly, a variety of somatic mutations have been discovered in histone genes [21,125]. It is worth noting that, independent of the number of mutated histone genes, the effect of mutation is dominant in inducing transformation, very often in children and adolescents [21,126,127]. Actually, there are features that clearly distinguish adult from pediatric gliomas, and one of them is the mutation of proteins related to the chromatin structure [128].

Now, cancer cells are clearly characterized by modification of their metabolism: in particular, they show an increase in lipid metabolism, thanks to which they can produce higher amounts of membranes in order to proliferate. Synthesis of higher amounts of lipids, in turn, requires reduced nicotinamide adenine dinucleotide phosphate (NADPH). Indeed, it was recently demonstrated the connection between breast cancer metastasis and the H3.3-driven upregulation of NADK enzyme that, in turn, favors NAPDH production, thus sustaining proliferation [129]. Another metabolic property of cancer cells is a high methionine requirement, especially for the ability of the adenosyl-derivative of methionine (S-Adenosyl methionine, SAM) to be used as a cofactor in both DNA and histone methylation [130].

A mutation of both H3.3 and H3.1 histones has been found in diffuse intrinsic pontine glioma (DIPG), a very aggressive tumor with a median overall survival of 8–11 months. This kind of tumor belongs to the category of diffuse midline gliomas (DMG) and primarily occurs in children. The lysine to methionine mutation at position 27 (K27M) of the H3.3 histone, in particular, is often associated with TP53 and ATRX mutations [131]. These latter mutations have been mainly found in midline brain structures, such as the thalamus, pons, and brainstem, where they associate with the alteration of the normal post-translational modification pattern of the H3 histone. The following variation in gene expression often leads to tumor development, even though it is not yet clear how these events are connected. One interesting finding is that the H3.3 K27M mutation differs from H3.1 K27M both for the association to secondary mutation and for the age of tumor onset [127,132,133,134]. Comparing two cell lines, one of which bore K27M mutation, Lewis and colleagues demonstrated that the mutated cell line presents a specific region of accessible chromatin, containing genes involved in neurogenesis and neuronal development, like the Achaete–Scute Family BHLH transcription factor 1 (ASCL1) and NeuroD [133].

One of the suggested H3.3 interactors is polycomb repressive complex 2 (PRC2), a methyltransferase involved in the control of gene expression and, more precisely, in the silencing of the promoters of specific genes by the trimethylation of lysine 27 of the H3 histones. In particular, in vitro studies showed binding between PRC2 and the mutated H3.3 [135,136]. The effect of the interaction between the H3.3 K27M mutated protein and the PRC2 complex is inhibition in the activity of the complex itself: the enhancer of the zeste homolog 2 (EZH2) component, a methyltransferase, stops methylating the other H3.3 histone residues in adjacent nucleosomes, causing general hypomethylation [31,136]. On the other hand, the interaction has not been proven in vivo, and the localization of the two proteins does not seem to coincide in the nucleus [137].

Another frequent mutation found in histone H3.3 in gliomas is the glycine to arginine (or valine) variation at position 34 (G34R/V). It has been suggested that this mutation affects the activity of the enzymes that modify the status of lysine 36 methylation, but a clear mechanism deriving from the original defect is not yet known. Differently from the K27M mutation, which can affect both the H3.3 and H3.1 histones, the G34R mutation is specific to the variant H3.3 [127].

As already mentioned, ATRX is a partner of the DAXX chaperone and forms with it a complex that deposits H3.3 on chromatin. A mutation to the components of this complex is frequent in different cancers and is associated with the pathway called alternative lengthening of telomeres (ALT) [138].

In many pediatric high-grade gliomas, ATRX is mutated, and, in the majority of cases, the evidenced mutation is associated with either the H3.3 K27M or H3.3 G34R/V mutation [132].

Moreover, the G34R/V mutation causes a reduced capacity of DNA repair in patients suffering from pHGG (high-grade glioma), leading to an enhanced susceptibility to DNA damage, also from radiotherapy [139].

In the alveolar rhabdomyosarcoma, H3.3 is overexpressed both in vivo and in cell lines: in this latter system, knockdown of the H3F3A gene reduces the ability of the cell to migrate, suggesting that the H3.3 variant regulates genes involved in motility, like melanoma cell adhesion molecule (MCAM, also known as CD146) [140].

Mutations of the H3.3 histone were also described in bone and cartilage cancers, such as the giant cell tumor (GCT) of bone [141]. One H3.3 mutation, changing lysine at position 36 with methionine (K36M), is associated with chondroblastoma, another tumor occurring especially in childhood [142]. The mutated H3.3 protein seems to block the activity of the multiple myeloma SET domain (MMSET) and SET domain containing 2 (SETD2) methyltransferases, altering the methylation landscape and gene expression and thus leading to neoplastic transformation [143].

## 6. Conclusions and Perspectives

In conclusion, all the cells of an organism are able to store memory of their past experiences in the form of specific modifications to chromatin structural organization. On the other hand, nerve cells not only store information related to the steps that allowed their development and differentiation, but they also store information related to events that determine macroscopic phenomena known as learning and memory. However, at the molecular level, the basal mechanisms that allow cognitive acquisition are very similar to the ones used by all cells to remember their history. Indeed, DNA methylation, histone PTMs of many kinds, and histone variant loading into chromatin constitute a collection of marks that is able to index the different genes to be transcribed. All these events, as discussed, are of central importance both for the general and specific structural organization of genes, and they also have an effect because of the existing interactions among different marks, as well as the binding of further factors (the readers), which are able to recognize the marks. Most importantly, long-term potentiation relies on these kinds of events, and it is induced mostly in the periphery of the nerve cells, as it depends on the signals that arrive through a variety of mechanisms, such as wiring (synaptic) and volume transmission (probably also mediated by extracellular vesicles) [100]. Therefore, we propose that a central aspect of all these events is the regulation of peripheral translation of mRNAs that encode proteins, which are then able to reach the nucleus, inducing adaptive modifications of nuclear chromatin. Pre-localization of mRNA is indeed an important aspect of adaptation in the nervous system, and the proteins that are able to bind mRNAs and transport/localize them play a key role both in neurodevelopment and the functions of the adult brain [144]. Moreover, RBPs also interact with non-coding RNAs, which are also part of the network that is able to regulate cell phenotype at the epigenetic level [144]. As a demonstration of the central epigenetic importance of RBPs in the nervous system functions, we finally cite the fact that, in almost all the neurodegenerative pathologies, alteration to the function/localization of RBPs has been evidenced [144].

## Figures and Tables

**Figure 1 ijms-24-11028-f001:**
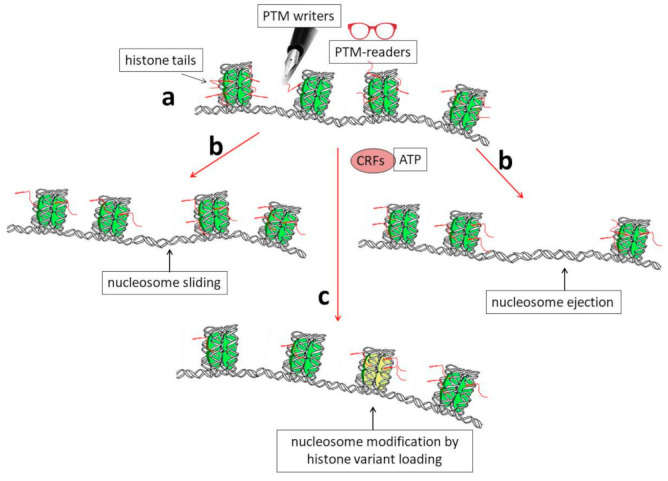
Schematic representation showing the biochemical mechanisms that allow dynamic modifications of chromatin organization. (a) Histone tails protrude from nucleosomes (red segments) and can be the targets of enzymes (PTM writers) that are able to introduce into them post-translational modifications (PTMs). These modifications can, in turn, directly affect DNA–histone interactions and/or allow chromatin structural modifications through the binding of other specific factors (PTM readers). (b) A modification of the nucleosome position by sliding or nucleosome ejection can be catalyzed by ATP-dependent chromatin-remodeling factors (CRFs). (c) Finally, the incorporation of histone variants into chromatin, such as H3.3, can induce structural and functional modifications of nucleosomes (shown as the yellow nucleosome core particle in the figure).

## Data Availability

The cited data are from bibliography and are thus all available in the literature.

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
