# Peer review of "Involvement of the H3.3 Histone Variant in the Epigenetic Regulation of Gene Expression in the Nervous System, in Both Physiological and Pathological Conditions"

_ijms, 2023, doi:10.3390/ijms241311028_

Round 1
Reviewer 1 Report
In the review entitled “The role of the histone variant H3.3 as an epigenetic regulator in the mammalian brain in both physiological and pathological conditions.” Authors discuss the role of the H3.3 histone variant, with particular attention to its production and loading into chromatin in the mammalian nervous system, both in physiological and pathological conditions. The review, although very short, include very important point of discussion, strongly underlying the crucial role played by Histone variants in controlling gene expression.
Author Response
We thank Reviewer 1 for the positive comments and for the time dedicated to our review.
Reviewer 2 Report
The authors review the role of H3.3 in the mammalian brain in normal and disease states. This is an informative review. However, more illustrations would enhance this review.
It was not clear what where the different functions of H3.3 in heterochromatin versus transcriptionally active chromatin. The authors did discuss how H3.3 with H2A.Z results in nucleosome instability. But other than this, the authors should provide more content on what is the role of variant H3.3. In preparation for the revised text, the authors need to read and comment on PMID: 28934504.
Recommended edits
The title should be revised. The authors do not discuss the role of H3.3.
The Introduction needs work to be better organized.
Line 81 - remove “anyway”
Line 91 – rewrite to be more direct “two genes is not to be related to the possibility to have two proteins with different activities,”
Line 220 – Note that H3.3 S31ph is with expressed genes (Armache, A., et al 2020. Histone H3.3 phosphorylation amplifies stimulation-induced transcription. Nature 583(7818): 852–857. doi:10.1038/S41586-020-2533-0.
Line 247 264 KAT not HAT
Missing references
The following references show that H3.3 is preferentially modified by “active marks”. The authors need to add the following papers and also do literature searches for other papers that show this.
PMID: 12086617 K. Ahmad, S. Henikoff, Mol. Cell 9, 1191–1200 (2002).
PMID: 19781938 A. Sakai, B. E. Schwartz, S. Goldstein, K. Ahmad, Curr. Biol. 19,
1816–1820 (2009).
H3.3 levels change with age
PMID: 28934504 H3.3 accumulation with age in neurons
The authors should also cite PMID: 37167393 as it is relevant to the review.
Sentences need to be re-written to provide more direct statements.
Author Response
First of all, we thank the Reviewer for the time dedicated to our review and for the advices.
Reviewer’s comments:
-The authors review the role of H3.3 in the mammalian brain in normal and disease states. This is an informative review. However, more illustrations would enhance this review.
As suggested, we added a new figure (Figure 1), that shows the main molecular mechanisms active in the cells in order to change chromatin structure and function;
- It was not clear what where the different functions of H3.3 in heterochromatin versus transcriptionally active chromatin. The authors did discuss how H3.3 with H2A.Z results in nucleosome instability. But other than this, the authors should provide more content on what is the role of variant H3.3. In preparation for the revised text, the authors need to read and comment on PMID: 28934504.
Actually, this point is still not completely clear in the literature; however, as suggested, we cited more papers and discussed, as possible, the implications of these observations;
- The title should be revised. The authors do not discuss the role of H3.3.
As suggested, the title was changed, and is now: “Involvement of the H3.3 histone variant in the epigenetic regulation of gene expression in the nervous system, in both physiological and pathological conditions”.
- The Introduction needs work to be better organized.
We modified Introduction, by adding, as mentioned above, a new figure, and by removing to a new paragraph (paragraph 2, entitled “General properties of genes encoding histone variants”), more specific information on the genes encoding H3.3;
- Recommended edits (lines 81, 91, 220, 247 of the previous edition):
All the suggested modifications, at the indicated lines, had been done;
- The following references show that H3.3 is preferentially modified by “active marks”. The authors need to add the following papers and also do literature searches for other papers that show this.
We added 9 further references, among which all the ones suggested by the Reviewer, and discussed these findings in the text.
Round 2
Reviewer 2 Report
accept